# Implementation and Validation of a Potential Model for a Moored Floating Cylinder under Waves

**Maria Gabriella Gaeta [1]**, **Giacomo Segurini [2]**, **Adrià M. Moreno [3]** and **Renata Archetti [1,*]**

[1]  Department of Civil, Chemical, Environmental, and Materials Engineering (DICAM), University of Bologna, 40136 Bologna, Italy; g.gaeta@unibo.it

[2]  INSA (S-lay, Lifting and Float Over Installation Analysis)—E&C OFFSHORE DIVISION, SAIPEM, 20097 San Donato Milanese, Italy; giacomosegurini94@icloud.com

[3]  Data and Reliability Dept., EOLOS Floating Lidar Solutions, 08110 Montcada i Reixac (Barcelona), Spain; adria.moreno.miquel@gmail.com

*  Correspondence: renata.archetti@unibo.it

**Abstract:** A three degrees-of-freedom model based on the potential flow theory was implemented to represent the motion of a slender cylindrical buoy under waves. The model calibration was performed by means of the comparison between the model results and the experiments performed at the Laboratory of Hydraulic Engineering of the University of Bologna (Italy). The dynamics of the floating cylinder, placed at the mid-section of the wave flume and anchored at the bottom through a mooring system of four catenaries, were obtained through videography analysis, providing surge, heave and pitch motions. The implementation of the mathematical model consisted of two main parts: The first has been developed in the frequency domain by applying NEMOH to assess the hydrodynamic coefficients of the object, i.e., the excitation, radiation and added mass coefficients; then, the used mooring system was included in the time-domain model, solving the motion of the floating cylinder, by calibrating the mooring coefficients by comparing the results with the data. The simplicity of the implemented model is a very important feature, and it should be used as a preliminary study to understand the response of moored floating cylinders and others floating bodies under waves.

**Keywords:** floating body; potential flow model; wave flume experiments; hydrodynamic coefficients; mooring system

## 1. Introduction

Interaction between water waves and floating objects has been extensively studied in naval, ocean and coastal engineering, and recently quite a remarkable amount of research [1–4] has been devoted to floating prototypes for wave energy conversion (WEC); this is due to the increasing investigations in renewable energy resources exploitation derived from the need for a new global energetic model, aiming to reduce fossil fuel consumption according to the IPPC [5].

The analysis of floating device responses to waves is often supported by numerical computations, generally based on Morison approaches, boundary element methods (BEM), computational fluid dynamics (CFD) or smooth particle hydrodynamics (SPH) models. The latter two approaches require high computational efforts [6–9] and are generally implemented after the preliminary design of the floating devices has been achieved.

Frequency-domain models, based on linearized potential flow theories, are often used in a proof-of-concept, hypothesized during the design phase of the floaters and under the assumptions of negligible fluid viscosity and nonlinear effects. This approach, largely adopted by marine engineering,

has been successfully used to simulate many interaction problems between non-breaking waves and floating bodies with dimensions much smaller than the wave length; among the others, the recent studies by [10–14] have been reviewed, together with the development of commercial codes, such OrcaFlex [15] and ANSYS Aqwa [16].

In case of floating bodies with a complex shape, the Boundary Element Method (BEM) is commonly implemented [17,18], obtaining steady solutions by solving numerically a boundary value problem in the frequency-domain. However, although these frequency-domain models are powerful and accurate in solving linear problems, they are usually implemented to provide hydrodynamic characteristics of floating devices and are coupled to time-domain simulations to analyze the response of moored structures [19–22].

Recently, Wendt et al. [23] performed a comparative analysis of modeling approaches for WEC devices, showing the similarity of results by using linear and (weakly and fully) nonlinear models under small and medium wave conditions.

The floating dynamics in the sea are also significantly influenced by the mooring restraints that have been recognized as important by several authors, e.g., [24–27]. The mooring system should be included during the first stages of the prototype development, modeling it in the dynamical analysis of the floating device, since the motion of the floater depends on the time- and position-dependent chain tension [26].

This study aims to present a properly developed three degrees-of-freedom (DoF) model, based on potential flow theory, in order to represent the motion of a floating body calibrated through the comparison with laboratory data. The implemented approach can be used as a validated tool to preliminarily optimize the design of complex bodies, as an innovative WEC prototype designed for the Mediterranean Sea states by the authors [28,29], and give a first estimate of their response, as well as in general provide an easy tool for the preliminary assessment of the dynamics of floaters under waves.

The paper also presents a set of data used to validate this model, which in the future will be used also to validate other codes (CFD). The scientific community has plenty of similar studies [23,27,30] since mooring modelling still need stages of calibration and validation, especially the laboratory-scale models.

In Section 2, the experiments performed at the Laboratory of Hydraulic Engineering of the University of Bologna (Italy) are described; the paper shows the implemented technique, based on videography, to analyze the dynamics of the moored object in the flume, providing surge, heave and pitch data motions. The mathematical model is described in Section 3, reporting the acting forces in the governing equation, accounting also for the catenaries mooring system and the implemented frequency-domain (Section 3.3) and time-domain (Section 3.4) models. The results of the calibration are presented in terms of hydrodynamic and mooring coefficients and the comparison with experimental data is discussed in Section 4. At the end, some conclusions close the paper, highlighting the implemented modelling approach as a simple and useful tool to support preliminary studies to design floating bodies and understand the response of a moored floating cylinder under waves.

## 2. Description of the Experiments

### 2.1. Experimental Set-Up and Generated Wave Conditions

The new experiments on floating body dynamics under waves were carried out in the wave flume of the University of Bologna. The flume, sketched in Figure 1, is 12 m long, 0.5 m wide and 1.0 m deep, and the waves are generated on the left-hand side by the vertical movement of a cuneiform-shaped piston-type wave-maker [31,32]. On the other side of the channel, a wave absorber panel is installed to reduce the wave reflection.

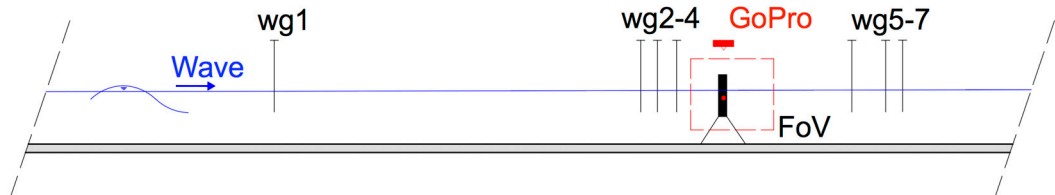

**Figure 1.** Longitudinal sketch of the wave flume at the Laboratory of University of Bologna (not at scale).

The floating object is a cylindrical and slender object made of plastic and lead: Figure 2 and Table 1 report the dimensions of the cylindrical buoy, with the indication of the centers of mass of the single parts and of the rigid body. While most of the hollow structure of the buoy is made of plastic, a lead block is placed at the bottom of the body in order to shift down the center of mass and allowing the buoy to maintain a vertical configuration while floating under waves.

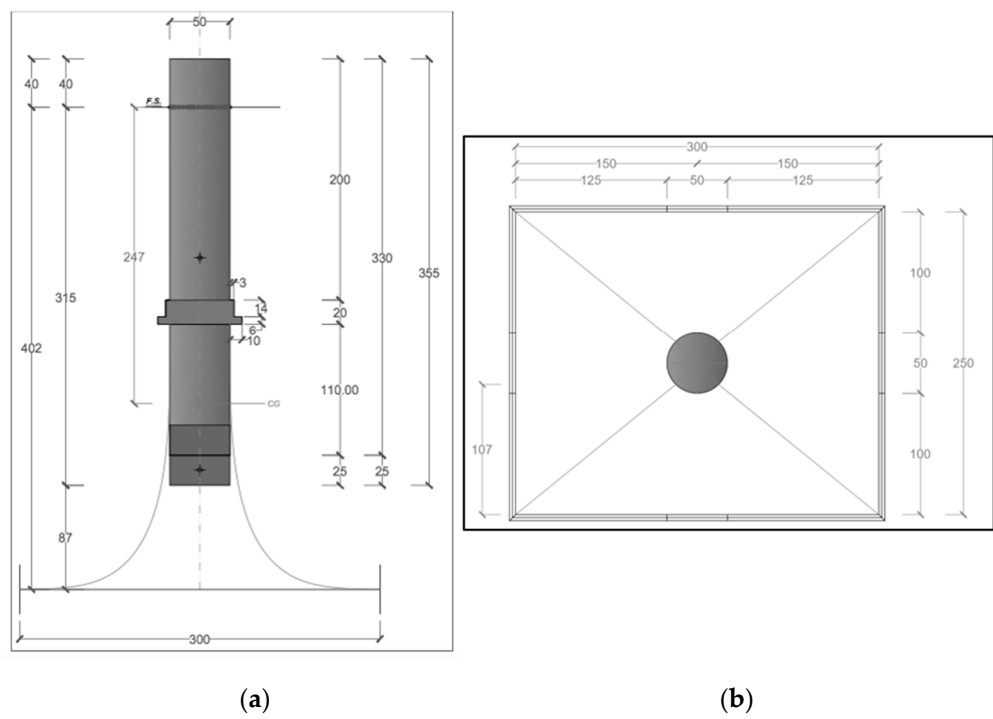

(**a**)                    (**b**)

**Figure 2.** Lateral (**a**) and top (**b**) views of the tested cylinder (measures in mm).

**Table 1.** Characteristics of the cylinder buoy and of the mooring system.

| Parameter | Cylinder |
| --- | --- |
| Height (m) | 0.355 |
| Diameter $D$ (m) | 0.050 |
| Total mass (kg) | 0.601 |
| Plastic mass (kg) | 0.188 |
| Lead mass (kg) | 0.413 |
| Position of center of gravity (m) | 0.247 |
| **Parameter** | **Mooring System** |
| No. of chains | 4 |
| Chain length (m) | 0.35 |
| Mass (g/m) | 19.5 |
| Nominal diameter (mm) | 0.95 |
| Material | Steel |

The mooring system of the cylinder (Figure 2) was made up of four steel chains characterized by a linear density equal to 19.5 g/m, corresponding to a chain nominal diameter of 0.95 mm. The chains were hooked to the buoy through a plastic crown placed 20 cm below the upper surface of the buoy.

The floating cylinder was located at the center of the flume as shown in Figure 1, moored at its equilibrium position, which was set at around 4.0 m away from the wave-maker.

Since the present tests were carried out in a wave flume, the multi-directionality of the response of the floating object to the waves could not be investigated for two main reasons: First, the implemented videography allows estimating the floating body motion only in the $x$–$z$ plane through the acquired planar images. In addition, the boundary effects of the walls could not be assumed completely negligible.

The choice of scale for the model was based on the specifications of the wave flume and on the wave conditions to replicate a representative Mediterranean Sea state, reducing the model according to Froude's law on a scale ratio of 1:64.

A recording system made of seven resistive-type wave gauges (represented in black in Figure 1) was distributed all along the channel, to acquire at a sampling frequency of 1 kHz the free surface and to reconstruct the incident, reflected and diffracted waves during the tests according to [33].

Furthermore, two GoPro cameras (in red in Figure 1) have been installed in order to record at 30 fps in full HD (1920 × 1024 pixel) the floating cylinder movements from the lateral and top sides of the view. The experiments were conducted in a dark environment, with two controlled light sources placed on the top and lateral sides of the flume in order to enhance the image contrast. The list of the experimental conditions is presented in Table 2.

**Table 2.** Test ID and values of $H$, $w$ and $\lambda$, $h/\lambda$ and $k_R$, characterizing the experimental conditions.

| Test ID | $H$ (m) | $w$ (rad/s) | $\lambda$ (m) | $k_R$ | $h\lambda$ |
|---------|---------|-------------|---------------|-------|------------|
| R03 | 0.007 | 8.72 | 0.80 | 0.15 | 0.500 |
| R04 | 0.008 | 8.16 | 0.90 | 0.15 | 0.444 |
| R05 | 0.009 | 7.66 | 1.00 | 0.16 | 0.400 |
| R06 | 0.010 | 7.39 | 1.10 | 0.18 | 0.364 |
| R07 | 0.010 | 6.98 | 1.20 | 0.20 | 0.333 |
| R08 | 0.011 | 6.68 | 1.30 | 0.22 | 0.308 |

At the beginning, a heave decay test was performed to get the natural frequency of the object, then a set of 6 regular waves was generated in the flume, with the aim to be later easily reproduced by a CFD model.

The reproduced waves were characterized by height $H$ in the range 0.006–0.030 m, frequency $w$ in the range 6.60–8.80 s and length $\lambda$ in the range 0.80–1.30 m. The water depth $h$ at the wave-maker was at maximum equal to 0.4 m and the reflection coefficients $k_R$ in the wave flume during the tests had reached values between 0.12 and 0.25. The generated waves keep being regular, and hence, linear wave theory was used to describe the waves themselves. Linear wave theory was used to distinguish shallow, deep and intermediate water situations for each test and then to study the particle velocity under the waves themselves. First the coefficient $h/\lambda$ was computed in order to verify which condition among the three previously exposed was satisfied: the resulted conditions are reported in Table 2, where the values are in the range 0.3–0.5. According to the application areas of wave theories as defined in [34], the reproduced waves primarily were in intermediate water depths with the exception of wave R03, propagating in a deep depth.

## 2.2. Video Analysis to Detect Body Dynamics

A videography was implemented to analyze the images acquired by the two GoPro cameras at the lateral and top views during laboratory tests. The implemented procedure largely adopted to estimate the floating object motion by [35–39], was developed in a Matlab environment, and consists of the following steps:

(a) Pre-processing, where lens distortion and 2D calibration were performed. The calibration process was performed by using the approach proposed by [40], where images (at least five) of a planar checkerboard placed in the mid-section of the flume were used. The definition of a conversion factor (from pixel to mm) was performed by taking as input one sample image up to reach a maximum conversion error less than 1 pixel (around < 0.3 mm). Finally, camera intrinsic and extrinsic parameters were provided as a calibration matrix, to obtain the correspondence between the image and the space (real) points. During each test, triggering for these cameras was performed using the GoPro application via Bluetooth. The method was verified by checking, both at rest and during the waves, the cylinder's dimensions, estimating its height and diameter, and resulting in a mean error of 4% and 9%, respectively;

(b) Adjustment of each image, aiming to enhance its quality, intensity and contrast in order to much more easily detect the target points on the surface. Each frame of the recorded videos was cropped in order to analyze a smaller area significantly, follow the object motion and reduce the computational times; and

(c) The final estimation of the cylinder motion, by means of the detection of control points on its surface. In particular, the image analysis consists of the detection of two markers on the cylinder, in order to follow the object translation (surge and heave motions) and rotation (pitch motion) during the tests. The first point is located at the buoy center of mass, not being influenced by rotation, while the second one belongs to the same axis of the first point, in order to estimate the object rigid rotation in the $x$–$z$ plane, by applying some easy trigonometric calculation.

Figure 3 shows the sequence of the implemented image processing: The cropped frame (a) is adjusted in contrast and transformed in a binary frame (b); then, the detection of the center of the markers (white pixels) is performed to follow the three DoF motion of the object (c).

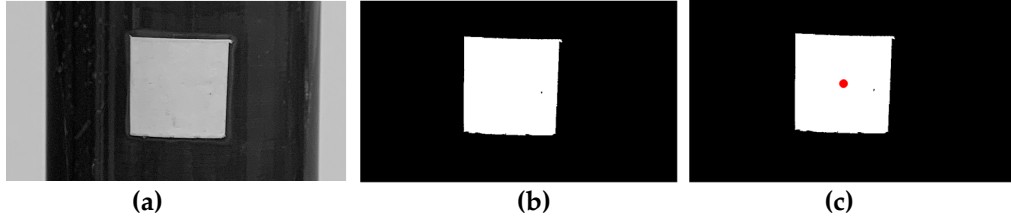

|      (a)      |      (b)      |      (c)      |

**Figure 3.** Sequence of the implemented image processing: original (cropped) frame (**a**), transformation into a binary frame (**b**), detection of the center of the marker (**c**).

The implemented image processing could be considered reliable to estimate the motion of the floating object in the $x$–$z$ plane, and the analysis of the top-side view camera was negligible sway and roll components in the cylinder response to the modelled waves.

## 3. Description of the Mathematical Model

A potential flow model was implemented to represent the buoy behavior under the different regular sea states, adopting three DoF, i.e., surge, heave and pitch. After a reminder of the acting forces for the studied case and the definition of the governing equation of the buoy cylinder, the implemented approach consisting of two main parts is presented. The first part is related to the frequency domain, where hydrodynamic coefficients were assessed. The second part of the model is related to the time domain, solving the buoy motion coupled to the mooring system in time under waves.

### 3.1. Wave-Induced Forces

Before describing the equations constituting the model, the acting forces on the floating cylinder during the waves are reviewed and their possible simplification is discussed according to well-known non-dimensional parameters.

Firstly, the Koulegan–Carpenter (KC) number was computed for each test condition, evaluating the ratio between drag and inertia forces. In case KC > 10, drag forces are predominant over inertial ones, while if KC ≤ 2 inertia forces are predominant with respect to drag forces and the latter can be neglected. As shown in Table 3, KC numbers takes intermediate values for the presented tests, getting bigger when wave height enhances. So, it is possible to conclude that both drag and inertial forces have to be considered in the modelling of the motion.

**Table 3.** Test IDs and values of KC and $D/\lambda$, characterizing the experimental conditions.

| Test ID | KC | $D/\lambda$ |
|---------|------|-------|
| R03 | 3.02 | 0.063 |
| R04 | 3.39 | 0.056 |
| R05 | 3.77 | 0.050 |
| R06 | 4.14 | 0.046 |
| R07 | 4.52 | 0.042 |
| R08 | 4.90 | 0.039 |

The influence of the diffraction forces was also investigated, estimating the ratio between the relevant buoy dimension, i.e., the cylinder's diameter $D$, and the wavelength $\lambda$. In case of $D/\lambda \ll 1$, no relevant diffraction phenomena occur, and hence diffraction forces can be neglected compared with the other acting forces. Table 3 shows the values of $D/\lambda < 1/5$ for the performed test, meaning that the cylinder diameter is small in comparison to the modeled wavelengths, and so diffraction forces can be neglected.

Finally, a summary of the acting forces and their mathematical expressions as implemented in the present potential flow model is given in Table 4: with index $j$ equal to 1 for surge, 3 for heave and 5 for pitch. The vector $S_j$ represents the object area, the matrices $R_{d,jk}$, $K_{jk}$, $f_{exc,j}$ and $C_{d,jk}$ include the hydrodynamic coefficients of radiation, hydrostatic stiffness, wave excitation and drag coefficients: the first three are computed in the frequency domain (see Section 3.3) and the latter coefficient has been obtained from the literature [41], resulting in (0.84; 0.98; 0.0) for surge, heave and pitch modes, respectively.

**Table 4.** Type of acting forces and their mathematical expression reported in the implemented potential model.

| Type of Force | Mathematical Expression | |
|---------------|-------------------------|---|
| Radiation damping force | $F_{r,j}(t) = \sum_{k=1,3,5} R_{d,jk}\, \dot{x}_k(t)$ | (1) |
| Drag force | $F_{d,j}(t) = \sum_{k=1,3,5} \frac{1}{2}\rho\, S_j\, C_{d,jk}\left|\dot{x}_k(t)\right|\dot{x}_k(t)$ | (2) |
| Hydrostatic restoring force | $F_{hyd,j}(t) = \sum_{k=1,3,5} K_{jk}\, x_k(t)$ | (3) |
| Excitation force | $F_{exc,j}(t) = f_{exc,j}\, \eta(t)$ | (4) |

The terms $x_k(t)$ and $\dot{x}_k(t)$ represent the buoy position and velocity, respectively, while $\eta(t)$ is the free surface elevation as experimentally recorded by the wave gauge closest to the floating object (wg4 in Figure 1).

Since regular waves were reproduced, the hydrodynamic coefficients maintain the same value both in the frequency and time domains.

The excitation force $F_{exc,j}$ is responsible for enhancing buoy motion, making the body oscillating far from its balance position; while all the others contribute as radiation, buoyancy and drag forces act as restoring ones, trying to bring the body back to equilibrium.

Like excitation force and radiation coefficients, an added mass matrix is frequency-dependent as well. It is summed to the inertia matrix in order to consider the in-motion equation of both mass coming from the body and the one associate with the displaced fluid.

### 3.2. Mooring System

In the laboratory, the floating cylinder was moored to the bottom through a system composed by four catenary lines, providing a strong symmetric response of the object with respect to the vertical plane. The chains were never stretched during the tests, since the generated incident waves were small enough to avoid the complete tensioning of the four lines.

Therefore, under this assumption, motion constrains due to the catenaries were modelled through the use of two different mechanisms by using linear quasi-static mooring stiffness. According to [42,43], restoring forces due to hydrodynamic damping induced by the lateral motion of the chains and to geometric stiffness were included in the model and reproduced as follows:

$$F_{damp,\,jk}\,(t) = K_{damp,jk}\,\dot{x}_k\,(t) \tag{5}$$

$$F_{stiff,\,jk}\,(t) = K_{stiff,jk}\,x_k(t) \tag{6}$$

where $K_{damp,jk}$ and $K_{stiff,jk}$ are the damping and stiffness coefficients, estimated through a calibration process by comparing for each DoF the model results with the experiments, globally accounting for the four chains.

Finally, the mooring effect was included in the governing equation of the motion, i.e., as the summation of the two forces acting in case the catenaries are not stretched:

$$F_{Moor,jk}\,(t) = \sum_{k=1,3,5} K_{damp,jk}\,\dot{x}_k\,(t) + K_{stiff,jk}\,x_k\,(t) \text{ with } j = 1,\,3,\,5 \tag{7}$$

### 3.3. Equations of Motion

With the previous assumptions, Morison's method allows to obtain water particles kinematic from the analytical solutions of different wave theories, assuming that the body does not disturb water particles motion, since the buoy is slender enough to not create any disturbance to the wave particles kinematic and diffraction forces, which are negligible compared with the inertia and drag ones.

The buoy was modeled as a rigid body moving under wave with three DoF: the horizontal translation along wave propagation direction (along the *x*-axis), the vertical translation (along the *z*-axis) and the rotation restrained in the vertical plane *x–z*.

According to Newton's second law and considering the previously defined assumptions and acting forces, the governing equations in the matrix form in time can be expressed as

$$\sum_{k=1,3,5}\Big[\big(I_{jk} + A_{jk}\big)\ddot{x}_k(t) + R_{d,jk}\,\dot{x}_k(t) + K_{damp,jk}\,\dot{x}_k(t) + \tfrac{1}{2}\rho\,S_{jk}\,C_{d,jk}\big|\dot{x}_k(t)\big|\cdot\dot{x}_k(t) \\ + K_{jk}\,x_k(t) + K_{stiff,jk}\,x_k(t)\Big] = F_{exc,j}(t) \text{ with } j = 1,\,3,\,5 \tag{8}$$

where $\ddot{x}_k$ stands as the acceleration vector in surge, heave and pitch modes, while $I_{jk}$ and $A_{jk}$ are the inertia and the added mass matrices, respectively.

### 3.4. Frequency-Domain Model

The hydrodynamic coefficients in Equations (1), (2) and (4) need to be estimated to calculate the acting forces in Equation (8). Since they are frequency-dependent, it results useful to calculate them in the frequency domain before passing to the time domain and the opensource NEMOH model [44] developed by Ecole Centrale de Nantes has been implemented to compute the first-order wave loads on the cylinder, in order to get added mass, radiation damping and excitation forces, by means of the Boundary Element Method (BEM).

The model solves the fluid velocity potential integration on the body surface, employing the method of Green's functions to transform a flow problem into a problem of source distribution on the body surface. The continuity of a Newtonian and inviscid fluid, characterized by homogeneity,

isotropy and initial rest conditions, atmospheric pressure above free surface and neglecting surface tension were assumed as hypotheses.

Before the BEM integration, the buoy surface was divided into elements in a 3D space (slices and panels) and a sensitivity analysis presented in the Appendix A was performed on the results, in order to get the more suitable mesh resolution, leading to $20 \times 700$ cells to discretize the tested cylinder (in Figure 4).

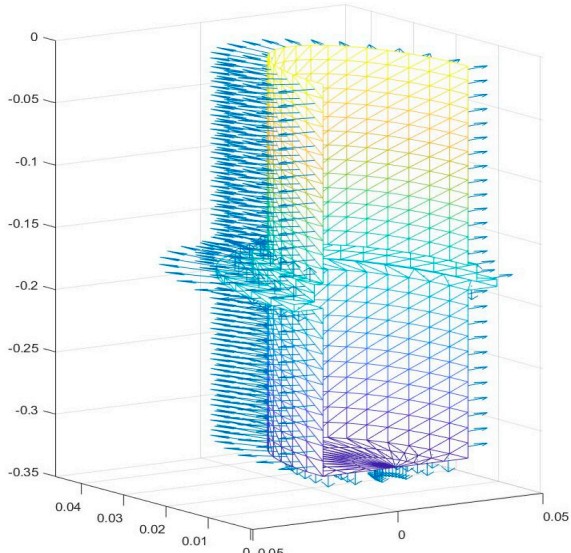

**Figure 4.** Computed mesh for the NEMOH model.

A frequencies vector $\omega$ was provided as a boundary condition in NEMOH, chosen in order to include both the tested wave frequencies and the natural resonance of the object, a body's characteristics depending on both mass and dimensions.

After the BEM integration, the added mass, radiation damping and excitation force coefficients were completely filled up and presented in Section 4.

Then, the Response Amplitude Operator (hereinafter RAO), representing first information about the motion amplitude according to different incident wave frequencies, was evaluated by applying the Fourier Transform function to the linearized equation of motion (Equation (8)), giving:

$$\text{RAO}_j = \frac{x_{jk}}{\eta} = \sum_{k=1, 3, 5} \frac{f_{exc,j}}{-\omega^2 \left(I_{jk} + A_{jk}\right) + i\omega R_{jk} + \left(K_{jk} + K_{stiff,jk}\right)} \tag{9}$$

with $j$ = 1, 3, 5 and where $\omega$ is the frequency of the wave exciting the buoy.

RAO is a parameter representative of the response of the floaters under the forcing of a unit wave height, and gives information on how the floating body moves in each DoF, giving a first view of the principal dynamical characteristics of a floater under waves.

### 3.5. Time-Domain Model

The implemented time-domain model solves Equation (8), where the input parameters can be divided into three categories. The first class is related to the environmental conditions, which are defined by the time series of the water surface elevation, as acquired from experiments at the wave gauge close to the cylinder, and the water particle velocities, as derived from the linear wave theory, according to the assumptions exposed in Section 2. The second input class refers to the device properties, consisting of its geometry and the characteristics of the installed mooring system. Finally, the last types of input are the so-called hydrodynamic coefficients, linking the wave conditions to the body response and defined in the previous frequency-domain model.

Equation (8) was solved as a first Order Differential Equation (ODE) by adopting the fourth order Runge-Kutta method as the most popular approximation because of its simplicity and efficiency [45]. A time step of 0.01 s was chosen according to the sampling frequency of the wave gauges and camera in the laboratory (i.e., 1 kHz and 30 Hz, respectively), so each wave period and cylinder motion was discretized by at least 100 points, as suggested by [45]. With a maximum number of iterations equal to 100, the solution was found with an accuracy of 1‰, which was assumed comparable to the one achieved in the laboratory data.

The simulations give as outputs the buoy position at the computed time step for each DoF, i.e., surge, heave and pitch.

## 4. Results and Discussion

### 4.1. Frequency-Domain Results

By adopting $20 \times 700$ mesh after the sensitivity analysis of the mesh, the hydrodynamic coefficients (Figure 5) were estimated by NEMOH in the range of the tested wave frequencies, i.e., between 6.6 and 10.0 rad/s.

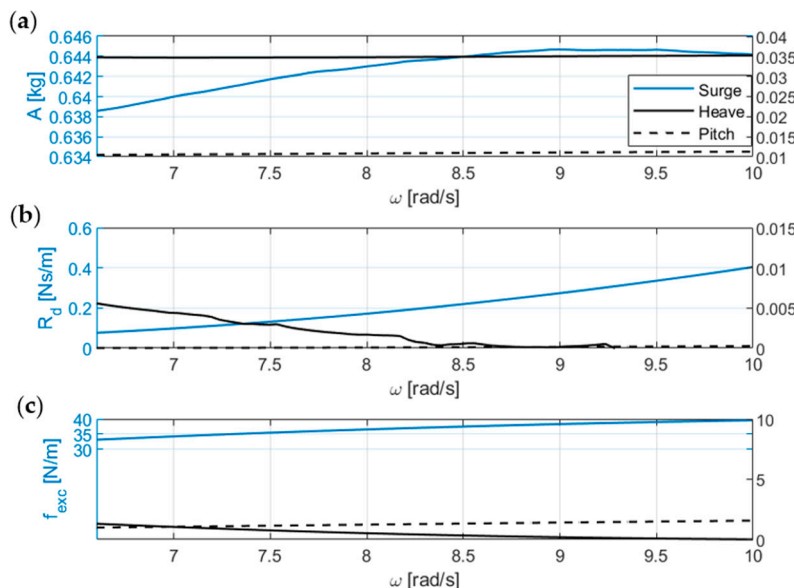

**Figure 5.** Added mass A (**a**), radiation damping $R_d$ (**b**) and excitation force $f_{exc}$ (**c**) coefficients for each DoF in the range 6.6–10 rad/s, as resulted by NEMOH.

The values of the hydrodynamic coefficients show that the added mass coefficients tend to maintain constant values throughout the frequency domain. Radiation coefficients tend to be a very small value at low frequencies; but, while the ones related to the heave and pitch ones keep being small, the surge coefficient increases. A possible reason could be related to the buoy dimension: Being a slender cylinder, influence areas are different along the horizontal and vertical sections with the latter being bigger than the one. Since the horizontal section is much more linked to surge motion, it can have a bigger influence on the free surface while perturbing it during the motion of the object. Between heave and pitch radiation coefficients, the latter maintains bigger values along the frequency domain, owing to the fact that pitch rotation is related with surge, while heave translation not. Excitation force coefficients do not reach constant values inside the frequency range of the tests.

*4.2. Time-Domain Results*

4.2.1. Free Heave Decay Test

A free heave decay test was carried in still water setting an unstable initial condition for the cylinder at $z_0 = -40$ mm, far away from the balance configuration (see also Figure 2). When simulation has begun, the buoy has started oscillating along the vertical direction, while any surge oscillation has arisen since no excitation forces, due to the waves, were provided.

The amplitude of the oscillation decreases until the balance configuration is assessed according to the natural object frequency: The heave decay in time for the numerical results and the laboratory data is illustrated in Figure 6.

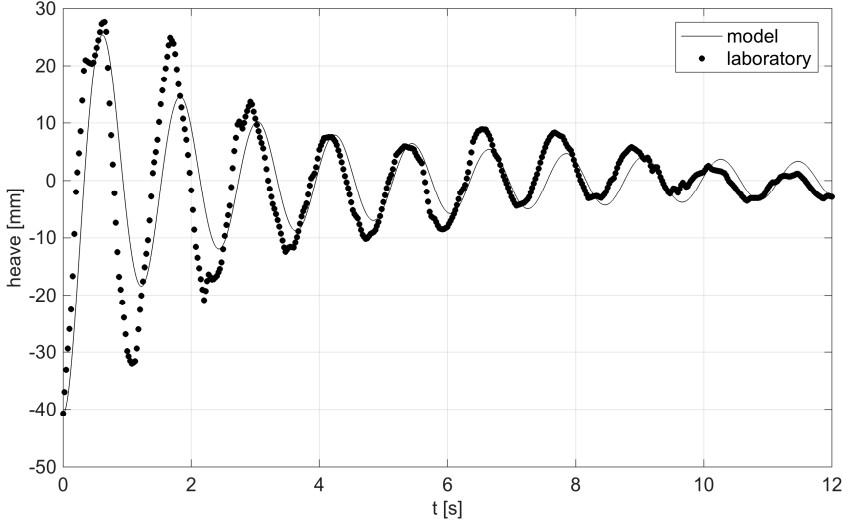

**Figure 6.** Free heave decay test: comparison between the model results and the laboratory data.

In the decay test, the structure moves at its natural frequency and the motion is dependent on the structure mass I, added mass A, radiation damping $R_d$ and hydrostatic stiffness K.

Crests and troughs of the heave decay signal in Figure 6 were detected and used to calculate the natural frequency $w_n$ as the average of the values and the first and second amplitude values $z_1$ and $z_2$. The results are shown in Table 5.

**Table 5.** Natural frequency $w_n$ and first and second oscillation amplitude $z_1$ and $z_2$: comparison between laboratory data and model results, with their relative errors.

| Parameter | Laboratory | Model/Error (%) |
|:---:|:---:|:---:|
| $w_n$ (rad/s) | 5.29 | 5.25/−0.75 |
| $z_1$ (mm) | 24.0 | 15/−37.0 |
| $z_2$ (mm) | 12.0 | 10/−15.0 |

The model results obtained for the case of free heave decay shows a good agreement with the laboratory data, reaching errors in the estimation of the natural frequency of around 0.75%, while for the computed first and second oscillation amplitudes, probably the absence of viscous effects in the model is responsible of higher discrepancies with the experiments, equal to 37% and 15%, respectively.

By considering the whole amount of acting forces, both linear and nonlinear, the analysis of the heave decay allowed catching the natural frequency of the object, which is related to its resonance achieved when the incident wave frequency matches the natural frequency of the body, producing the higher possible response.

### 4.2.2. Model Calibration and Response to Waves

Figures 7–9 show the surge, heave and pitch response of the floating cylinder during test R05 and the comparison between the laboratory data and the calibrated numerical results. Time series and spectrum of the motions are reported in the left and right panels.

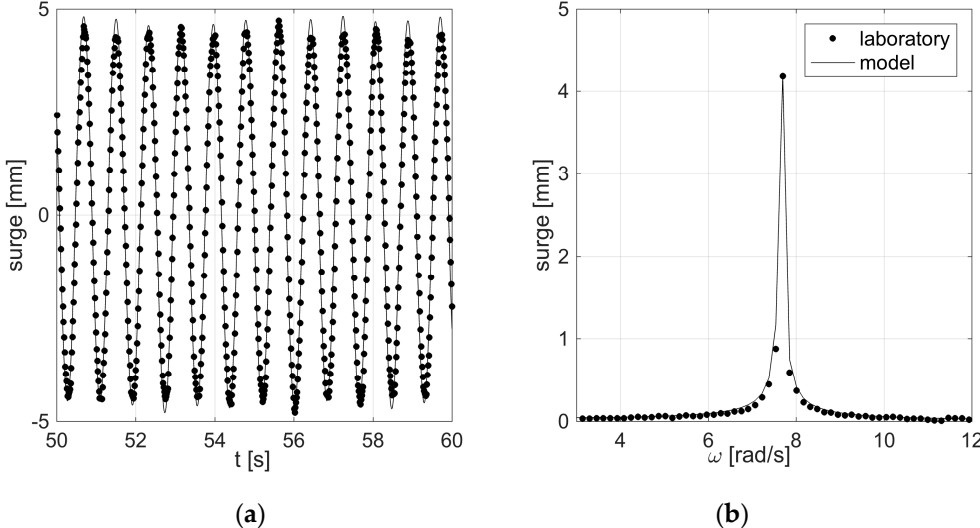

(**a**)                                                            (**b**)

**Figure 7.** Surge response during test R05: comparison between laboratory and model results of the time series of oscillation (**a**) and spectrum (**b**).

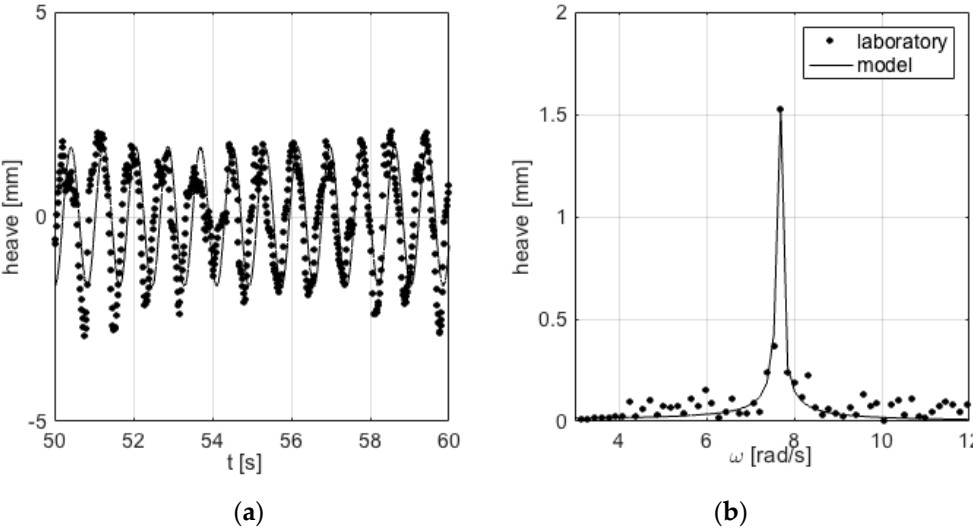

(**a**)                                                            (**b**)

**Figure 8.** Heave response during test R05: comparison between laboratory and model results of the time series of oscillation (**a**) and spectrum (**b**).

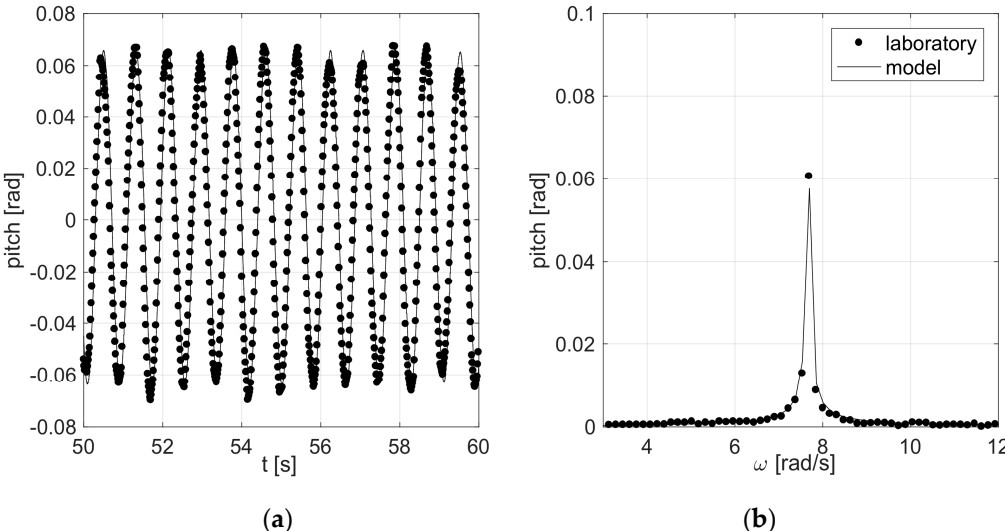

**Figure 9.** Pitch response during test R05: comparison between laboratory and model results of the time series of oscillation (**a**) and spectrum (**b**).

The final values of the mooring coefficients after calibration are plotted in Figure 10, where the damping and stiffness coefficients are reported for each DoF as included in the model.

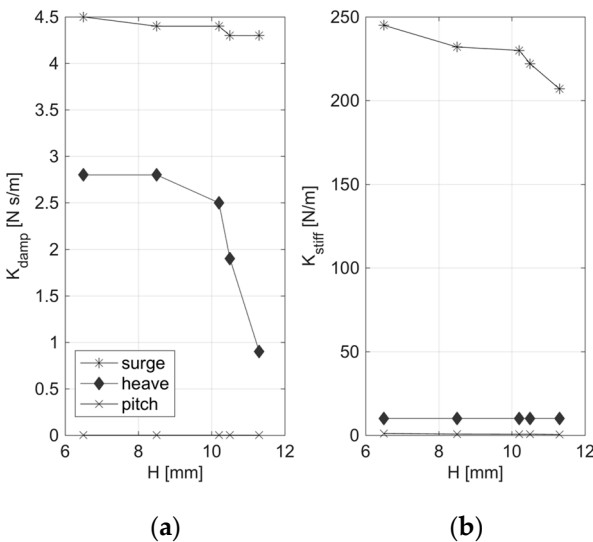

**Figure 10.** Calibrated values for the mooring system against the tested wave heights (R03-R08): damping (**a**) and stiffness (**b**) coefficients for each DoF.

The calibration against the experimental data was carried out in order to maximize as much as possible the values assumed by $K_{damp}$, that is responsible for a variation in the oscillation amplitude of the motion signals. Consequently, $K_{stiff}$ has been assessed, acting both on the oscillation amplitude and frequency.

The derived values of $K_{damp}$ are very low, in the range of 4.2–4.5 and 0.8–2.8 Ns/m, for surge and heave motions, respectively. They are seen to assume a decreasing trend, depending on the wave-induced forces on the body, that the mooring system must fight against and that are related to the displacements experienced by the chains.

The coefficients of $K_{stiff}$ present higher values in case of surge motion, in the range of 200–250 N/m, slightly decreasing with the wave heights, while for heave, their values remain constant and small (approx. 10 N/m).

The pitch rotation is not as influenced by the mooring system as surge and heave translations, and this issue is shown in the found calibration coefficients, equal to zero and below one for $K_{damp}$ and $K_{stiff}$, respectively. Finally, under the tested conditions, the forces exploited by the implemented mooring system have been found different along the three DoF, resulting higher for the surge motion, and reaching the smallest values for pitch.

The final values of the calibrated mooring coefficients have achieved discrepancies between the model results and the laboratory data in the range of 4% and 1% in terms of motion amplitudes and frequency, respectively, reaching satisfying outcomes for the study purposes.

In Figure 11, the resulted RAOs for the tested conditions are shown together with the NEMOH computations in absence of mooring: values for surge, heave and pitch are plotted in the frequency range of 6.6–10 rad/sec. The results show how the catenary mooring system has had a considerable contribution to the surge and pitch modes, leading the cylinder response independently to waves the in $x$ translation and $x$–$z$ rotation, instead providing for less effects on the heave RAO, where the test results follow the trend by NEMOH computations.

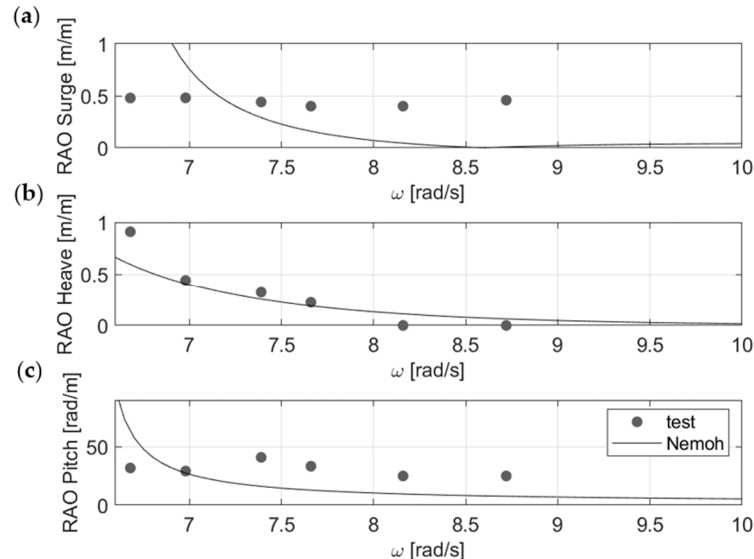

**Figure 11.** RAOs for the three DoF as resulted from the experiments (dots) and by NEMOH computations in absence of mooring (solid line). Surge (**a**), heave (**b**), and pitch (**c**).

## 5. Conclusions

In the present paper, a mathematical model based on the potential flow theory was implemented in order to represent the dynamical response of floating objects under regular waves. The model was calibrated by comparing the numerical results and the experiments performed at the Laboratory of Hydraulic Engineering of the University of Bologna (Italy). The global dynamic response of the floating cylinder, anchored at the bottom through four catenaries, were obtained by implementing a videography analysis, which has provided surge, heave and pitch motions under the tested regular waves. In the frequency domain by applying NEMOH, the hydrodynamic coefficients of the governing motion equation was assessed, i.e., excitation, radiation and added mass coefficients, and then coupled to the governing equations of the object motion in the time domain, where the mooring system was included by calibrating the damping and stiffness coefficients with the laboratory data, which can be made available to the readers.

The implemented model represents a validated tool to understand the global dynamic response of a floating body under waves both in the frequency and time domain. Tests with the same wave forcing and mooring conditions will be performed to study the dynamics of innovative WEC devices with a similar shape and mass of the studied cylinder, and the presented model will be used to perform a

preliminary analysis on the device performance. Additionally, the validated mooring coefficients will be also used to implement CFD simulations and validate analytical models of catenaries.

**Author Contributions:** Conceptualization, M.G.G., G.S., A.M.M. and R.A.; methodology, G.S., R.A. and A.M.M.; numerical model, G.S., and A.M.M.; validation, M.G.G. and G.S.; formal analysis, G.S. and M.G.G.; laboratory investigation, M.G.G. and R.A.; writing—original draft preparation, M.G.G. and G.S.; writing—review and editing, R.A.; supervision, R.A. All authors have read and agreed to the published version of the manuscript.

**Funding:** This research received no external funding.

**Acknowledgments:** Agnese Paci's contribution in the laboratory investigation is kindly acknowledged.

**Conflicts of Interest:** The authors declare no conflict of interest.

## Appendix A

A sensitivity analysis was performed varying the size of mesh elements through a variation of the number of total panels and comparing RAO values in the three DoF for the different resolutions in order to achieve better and stable results. An angular discretization was fixed to a number of slices equal to 20. Besides, an increasing number of total panels was investigated in the analysis and the mesh resolutions are reported in Table A1.

**Table A1.** Characteristics of the different meshes considered for the sensitivity analysis, with a fixed number of slices equal to 20.

| Test ID | N. Panels | Cell Dimension (cm$^2$) |
|---------|-----------|-------------------------|
| B1 | 100 | $0.92 \times 7.00$ |
| B2 | 200 | $0.92 \times 3.11$ |
| B3 | 300 | $0.92 \times 2.15$ |
| B4 | 400 | $0.92 \times 1.65$ |
| B5 | 500 | $0.92 \times 1.33$ |
| B6 | 600 | $0.92 \times 1.12$ |
| B7 | 700 | $0.92 \times 0.96$ |
| B8 | 800 | $0.92 \times 0.82$ |

Eight different grids, from a coarse one to a more refined one, were tested, and RAO values have been computed for each configuration. Figures A1–A3 show surge, heave and pitch RAO respectively for different number of elements.

With an increasing number of total panels, a much more precise solution is achieved, and RAO values converge up to a stable and unique solution for the three DoF. Finally, a mesh of $20 \times 700$ cells was chosen to perform the frequency-domain analysis of the object motion under waves.

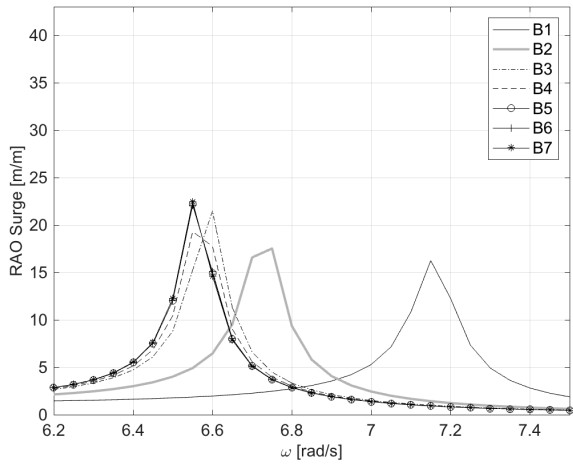

**Figure A1.** Comparison of surge RAO for sensitivity analysis.

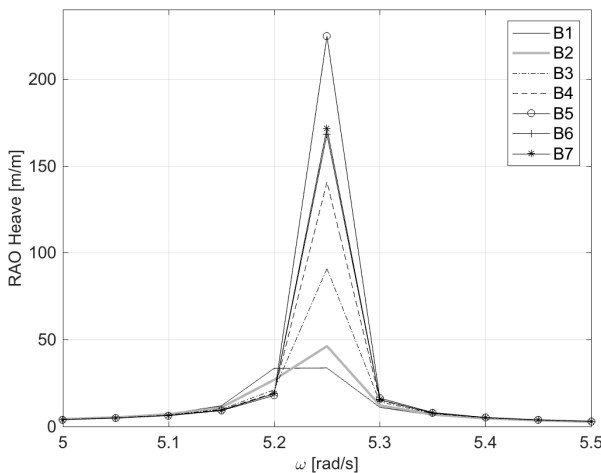

**Figure A2.** Comparison of heave RAO for sensitivity analysis.

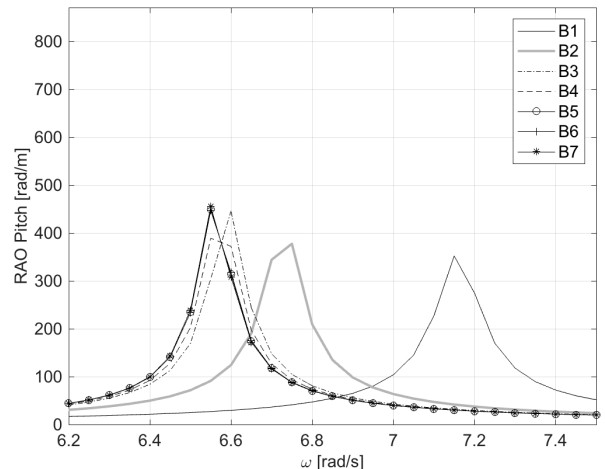

**Figure A3.** Comparison of pitch RAO for sensitivity analysis.

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
