# Peer review of "Implementation and Validation of a Potential Model for a Moored Floating Cylinder under Waves"

_jmse, doi:10.3390/jmse8020131_

Round 1

Reviewer 1 Report

The present paper describes a number of tests and a mathematical model for simulating motions of a WEC. The paper needs English editing and a re-structuring. The purpose of the paper is not clear to me, and I think a more scientific approach should be applied. The novelty and contribution of this paper needs to be addressed.

My comments are listed below:

Line 38: Inconsistency in citation style.

Line 39-47: Many studies have considered dynamic tools for WECs e.g.

Mathematical modelling of Mooring Systems for Wave Energy Converters”, J. Davidson & J. Ringwood. -“Screening of Available Tools for Dynamic Mooring Analysis of Large Wave Energy Converters”, J.B. Thomsen, F. Ferri & J.P. Kofoed.

Line 50-53: Do you mean that fully non-linear simulations like CFD is used or time-domain simulations using frequency-domain parameters.

Line 55-57: Other studies have also treated the applicability of models based on experimental data – these could be cited. E.g.:

“Numerical Model Validation for Mooring Systems: Method and application for WECs” Harnois et al. “validation of a Tool for the Initial Dynamic Design of Mooring Systems for Large WECs”, Thomsen et al.

Line 67: How do the authors define a reliable tool. No sources of errors, scalings uncertainties etc. is described in the paper, meaning that it is not reliable.

Line 69: Something is wrong in the citation.

Line 68: Is the Mediterranian only interesting?

Line 72: It could be beneficial to cite other studies that has done similar experimental tests:

“Experimental testing of Moorings for Large WECs”, Thomsen et al.

Introduction: English language needs checking. The novelty and the purpose of the paper is not clear. What is the contribution to the scientific society and why is the study relevant. A bit more introduction to the WEC is necessary. Is mooring the focus or modelling of the WEC?

Line 92-97: Does the model resemble any full-scale WEC?

Line 98: It is stated that the catenary chains has negligible weight. I do not understand this statement, since the restoring force in a mooring of catenary chains are resulting from the weight of the lines. Does the mooring provide any stiffness and station-keeping ability, if the weight is negligible?

Figure 2: Since mooring appears to be important for the paper, it would be beneficial to have more detail of the mooring as well and a top-view of the flume, mooring and model.

Line 137: Why is only regular waves tested?

Line 143 and Table 1: How is it possible to see if the is deep water waves or not. The limit of deep, intermediate and shallow water should be stated.

Table 1: Lambda is not defined.

Line 156-159: Other studies have investigated other methodologies for motion caption:

“Optical non-contact floating object tracking using an open-source library”, Ferri et al. “Study of Mooring Systems for Offshore Wave Energy Converters”, Paredes

Line 189: How does a mooring with four lines prevent motion in the y-axis? I guess that maybe radiation etc. can cause minor waves across the flume, giving some motion in the y-axis.

Section 3.2 Mooring System: Why is only two lines used in the mathematical model? Isn’t the purpose of the study to validate whether a model can be made, which provides similar results to the experimental work. If the authors apply a completely other mooring in the mathematical model, I don’t think the work provides any useful results.

Line 241-246: I do not understand the description of why a mooring with two lines are suitable.

Line 258-260: I do not understand, why a different mooring is applied and then tuned to resemble the experimental data. I do not think it is a validation of a model. The authors merely tune all input to fit the experiments and then compare them to the experiments. In that case resemblance will be present. What if no experiments are available or another mooring is desired in full-scale. Then it is unknown how the model behaves.

Line 276-277: Index has been used earlier without definition. It should be stated earlier.

Line 299: Now BEM is used even though it was stated earlier that diffraction is neglected?

Line 310: I do not understand what the frequency vector is?

Line 345: What does the choice of integration scheme, number of iterations and time step mean for the results.

Line 356-357: Does the calculated coefficients make sense?

Figure 5: It seems like the discretization of frequencies are very coarse. I think that the peak frequency could change if a finer discretization was used. I also think, that the RAOs are a bit unrealistic. A 1m amplitude wave will result in 40m surge, 200m heave and 900rad pitch. It seems top high.

Figure 6: I think the results look strange. There are no added mass in heave?

Line 392: How is “good agreement” defiend.

Chapter 5: I think the section/chapter is confusing and need a proper structure. What is the purpose? There are no discussion of results and whether they are realistic or not. I do not understand how the model can be used in full-scale. There are no discussion of sources of error.

Line 447-448: It is stated that the structure was tested under sea states resembling the Mediterranean Sea. I do not agree with this statement as only regular waves were tested.

Line 459: I do not think, that the model is a valid tool. It required tuning of all parameters in order to resemble experiments. If other moorings is used, the model cannot be used.

Conclusion: The purpose of the paper is still not clear to me. It needs to be highlighted and the paper should be structured to better resemble the purpose.

Author Response

Please find the point by point response in the attached file

Reviewer 2 Report

Review for JMSE-693091:  Implementation and validation of a potential model for a moored floating cylinder under waves

Main comments:

The manuscript conducted intensive lab experiments and developed a numerical model

 to represent the motion of a cylindrical buoy under regular waves. The manuscript was well written with high merit. It is ready for publish in the current version.

Author Response

Dear Review,

Thank you for your comments and for the high appreciation of the paper. 

The paper was in a large part rewritten, and improved considered the comments made by the others two reviewers.

Reviewer 3 Report

The paper presents laboratory experiments and 3 DoF model results for cylindrical mooring system. This paper is very interesting because of the it presents a simple modelling approach that has a strong potential application for quantifying the dynamics of floating wave energy conversion devices. The work modeled the motion of moored cylinder under wave action using the potential flow approximation and compared/calibrated with laboratory experiments. The presentation of the paper is clear but the readers who need to know:

how and what is novel about the method that warrant this paper (please clarify this in the abstract, introduction and the conclusion): is there an implementation in NEMOH that does not allow incorporation of mooring constraint? The gap is currently not clear! Whats the influence of parameters such as the aspect ratio of the cylinder, mooring length and stiffness  on the dynamics under the specified wave condition. Although, the work varied the wave properties and the selections are well justified, currently the paper uses one cylinder aspect ratio and mooring properties.   The discussion should include how the approach and the configuration can be extended to other practical applications.

Minor comments is on some occasional grammatical errors and omission in the presentation, I will suggest a further proofreading of the manuscript. Few of those are:

Line 69, Authors [10,?]

Line 131 planned should be removed

Lines 156 - 157 should be rephrased

Line 208 ... inertia forces not ....inertia ones

Line 231 obtained from literature?

Equations 8 and 9 are missing

Line 395 reference to a non-existing Equation 9

Line 401 Natural frequency of the body?

Author Response

(The authors gave the same response as above.)

Round 2

Reviewer 1 Report

The authors have answered the comments to the paper thoroughly, and the paper is ready for publication after editorial review.